# Focus of Ongoing Onchocerciasis Transmission Close to Bangui, Central African Republic

**DOI:** 10.3390/pathogens9050337

**Published:** 2020-04-30

**Authors:** Eric de Smet, Salvatore Metanmo, Pascal Mbelesso, Benoit Kemata, Joseph N. Siewe Fodjo, Farid Boumédiène, Hilda T. Ekwoge, Emmanuel Yangatimbi, Daniel Ajzenberg, Orphee Badibanga, Pierre-Marie Preux, Robert Colebunders

**Affiliations:** 1Global Health Institute, University of Antwerp, 2610 Antwerp, Belgium; eric.desmet@zol.be (E.d.S.); JosephNelson.SieweFodjo@uantwerpen.be (J.N.S.F.); 2INSERM, University of Limoges, CHU Limoges, IRD, U1094 Tropical Neuroepidemiology, Institute of Epidemiology and Tropical Neurology, GEIST, 87000 Limoges, France; salvatore.metanmo@gmail.com (S.M.); farid.boumediene@unilim.fr (F.B.); ajz@unilim.fr (D.A.); pierre-marie.preux@unilim.fr (P.-M.P.); 3Faculté des Sciences de la Santé, Université de Bangui, Bangui BP:3183, Central African Republic; pmbelesso@yahoo.com (P.M.); yang_emma1@yahoo.fr (E.Y.); 4Onchocerciasis Control Programme, Neglected Disease Control Programme, Bangui P.O. Box 883, Central African Republic; bkemata@yahoo.fr; 5HILPharma Organization, Yaoundé P.O. Box 25625, Cameroon; hildaekwoge@gmail.com; 6Association to Promote Neurosciences (APRONES), Kinshasa XI P.O. Box 127, Democratic Republic of the Congo; orpheebadibanga@gmail.com

**Keywords:** onchocerciasis, epilepsy, Ov16 seroprevalence, children, Central African Republic

## Abstract

Recently, there were anecdotal reports of a high number of persons with epilepsy, including children with nodding seizures in the Landja Mboko area located about 9 km from the capital city Bangui, Central African Republic. We suspected the area to be endemic for onchocerciasis, and that the alleged increase in the number of epilepsy cases was due to ongoing *Onchocerca volvulus* transmission. However, ivermectin mass drug distribution (MDA) had never been implemented in the area. Therefore we performed an Ov16 antibody prevalence study among children, aged 6–9 years, using the biplex rapid diagnostic test (SD Bioline Oncho/LF biplex IgG4 RDT). The overall Ov16 seroprevalence was 8.9%, and that of lymphatic filariasis (LF) was 1.9%. Ov16 seropositivity was highest in Kodjo (20.0%), a village close to rapids on the river. Our study shows that there is ongoing *O. volvulus* transmission in the Landja Mboko area. We recommend that the extent of this onchocerciasis focus should be mapped, and the introduction of ivermectin MDA should be considered in these communities.

## 1. Introduction

Onchocerciasis, a filarial disease caused by *Onchocerca volvulus,* is a country-wide public health problem in the Central African Republic (CAR). A rapid assessment in 2006 revealed that onchocerciasis transmission was ongoing in most parts of the country, with about 1,601,059 persons residing in meso- or hyper-endemic areas [1]. Due to the armed conflict that started in 2013, the CAR is currently confronted with a serious humanitarian crisis. As a consequence, onchocerciasis elimination efforts have been interrupted in certain parts of the country. In 2018, the estimated number of individuals requiring ivermectin treatment was 2,688,483 but it was reported that only 921,480 persons were treated with ivermectin [2]. More recent findings by CAR’s Ministry of Health show that in 2019, the onchocerciasis endemicity mapping was not very different from the baseline assessment of 2006 (Figure 1). 

Based on the initial rapid epidemiological mapping of onchocerciasis (REMO) studies in CAR, only the north-western areas of the country were identified as being meso- or hyper-endemic for onchocerciasis and community-directed treatment with ivermectin (CDTI) was implemented in those areas in 1993. In 1994, a population-based survey of blindness and visual impairment was conducted in the district of Bossangoa (in the Ouham river valley) [3], which showed substantial onchocerciasis-related ocular problems in the area. Besides ocular manifestations, it was also suspected that onchocerciasis may be responsible for epilepsy in the Bossangoa area. A matched case-control study was therefore conducted in 1996, which found that 39.6% of the epileptics and 35.8% of the controls were infected with *O. volvulus;* no significant association was found between onchocerciasis and epilepsy (odds ratio = 1.21, 95% confidence interval 0.81–1.80) [4]. This lack of association may have been caused by previous ivermectin use in these populations, as demonstrated by previous studies [5,6]. The CAR is also considered to be endemic for lymphatic filariasis (LF) but there are no recent data about the prevalence of LF in the different parts of the country [7].

Recent anecdotal findings from CAR suggest a high burden of epilepsy, including nodding seizures, in the Landja Mboko area located about 9 km from the capital city Bangui. Although this area had previously been classified as being hypo-endemic for onchocerciasis and hence never benefitted from CDTI, we suspected that the increase in the number of epilepsy cases, particularly among children, was due to ongoing *O. volvulus* transmission. We therefore conducted community-based surveys to confirm this anecdotal epilepsy burden and also assess the onchocerciasis situation in 2020. This paper reports our preliminary findings regarding the current onchocerciasis transmission status in the villages of interest. 

## 2. Methods

The study was conducted in January 2020 in the villages of Landja Mboko, in the south western part of CAR, about 9 km from the capital city Bangui (Figure 2). 

Using a door-to-door approach, non-epileptic children aged 6–9 years of both sexes were recruited for an Ov16 seroprevalence study; the 5–9 years age group is indeed recommended by the World Health Organization (WHO) when assessing ongoing onchocerciasis transmission patterns [8]. We excluded children suffering from any known illness or for whom we did not obtain assent/parental consent to participate. The socio-demographic and anthropometric information of participants was noted, and they were all tested for Ov16 IgG4 antibodies using a biplex rapid diagnostic test (SD Bioline Oncho/LF biplex IgG4 RDT), which also measures exposure to lymphatic filariasis (LF). Using rigorous aseptic procedures, the participants were finger-pricked to obtain a drop of blood; the tests were performed as per the manufacturer’s instructions, and the results were noted for each participant.

## 3. Results

A total of 259 children aged 6–9 years from five villages were included in this study (Table 1); 135 (52.1%) of them were males. Participants in Kodjo and Landja 2 villages had significantly higher median ages (7.5 years), compared to the other study villages where the median age was 7.0 years (Kruskal Wallis *p*-value = 0.025). Only two (0.8%) participants reported ever taking ivermectin in the past, because they happened to be in CDTI areas during the ivermectin distribution period.

The overall Ov16 seroprevalence was 8.9%, and that of LF was 1.9% (Table 2). Ov16 seropositivity varied significantly across the villages, with the highest values in Kodjo village (20.0%), followed by Landja 1 (14.3%), Landja 2 (4.7%), Mangapou 2 (4.2%), and Belespoir (0%).

The median age of Ov16-positive participants was higher than that of Ov16-negative children, although the difference was not significant (8 years vs. 7 years; Mann–Whitney U *p*-value = 0.378). However, more Ov16-positive children were underweight than their Ov16-negative counterparts; *p* = 0.018 (Table 3). 

## 4. Discussion

Our results show that there is ongoing onchocerciasis and LF transmission in the Landja Mboko area of CAR. The Ov16 seropositivity rate of 8.9% in children < 10 years is far above the 1% threshold proposed by the WHO in deciding whether an area should be benefitting from mass treatment with ivermectin [8]. Given that this region was also mapped as being loiasis-endemic [9], care must be taken when deciding which onchocerciasis elimination strategies are to be implemented in such a setting to avoid severe adverse events due to ivermectin treatment [10]. Hence, alternative strategies such as test-and-not-treat and vector control approaches could be considered [8,11].

Particular attention must be given to Kodjo village, where children had a much higher Ov16 seroprevalence than in all other study sites. Compared to other villages, Kodjo is situated only 200 m away from river Oubangui which most likely constitutes a suitable breeding ground for blackflies (the vector that transmits the onchocerciasis parasite). Previous research has documented an increasing burden of not only onchocerciasis, but also epilepsy, as the distance from the breeding ground decreases [12]. A similar phenomenon may explain the anecdotal reports of frequent epilepsy in this area. The findings from the door-to-door epilepsy survey in this same area will be published separately and may indeed shed more light on the current situation in Landja Mboko.

In our study, we observed that Ov16-positive children reported more frequent itching and were more often underweight compared to those who tested negative. A similar trend was observed when assessing the frequency of stunting in both groups, albeit being statistically non-significant. While retarded growth has previously been associated with onchocerciasis [13], it is unclear from this study whether the low anthropometric parameters observed in our participants were due to nutritional problems or exposure to *O. volvulus.*


A limitation of this study is the fact that only Ov16 rapid tests were used to assess onchocerciasis transmission, without confirmation using ELISA (the gold standard technique for Ov16 antibody detection, more sensitive than rapid tests) nor complementing our research with entomology. Our study area is also considered to be endemic for loiasis [9], but we did not determine the prevalence of *Loa loa* infection in our study population. While LF detection using RDT may suffer from cross-reactions with *Loa loa* [14], the performance of the Ov16 IgG rapid test in detecting exposure to *O. volvulus* is very satisfactory: sensitivity of 89.1% (95% CI: 86.2%–92.0%) and specificity of 97% (95% CI: 95.4%–98.6%) [15]. Our results therefore provide compelling evidence that Landja Mboko constitutes a hitherto ignored onchocerciasis focus in the CAR, requiring urgent attention from the public health authorities. 

## 5. Conclusions

Our data shows that there is ongoing *O. volvulus* transmission in the Landja Mboko area, where there is co-endemicity with loiasis. We also noted the occurrence of LF, although to a much lesser extent than onchocerciasis. Therefore a more in-depth re-assessment of the onchocerciasis situation (via skin snip surveys and entomological studies) in this area is warranted. Given the risk of encephalopathy when introducing ivermectin mass drug administration in loiasis-endemic areas, the prevalence and intensity of *L. loa* infection also needs to be assessed in these villages. In the light of the paradigm shift from control to elimination of onchocerciasis, it is crucial that all transmission foci be identified and appropriate interventions initiated.

## Figures and Tables

**Figure 1 pathogens-09-00337-f001:**
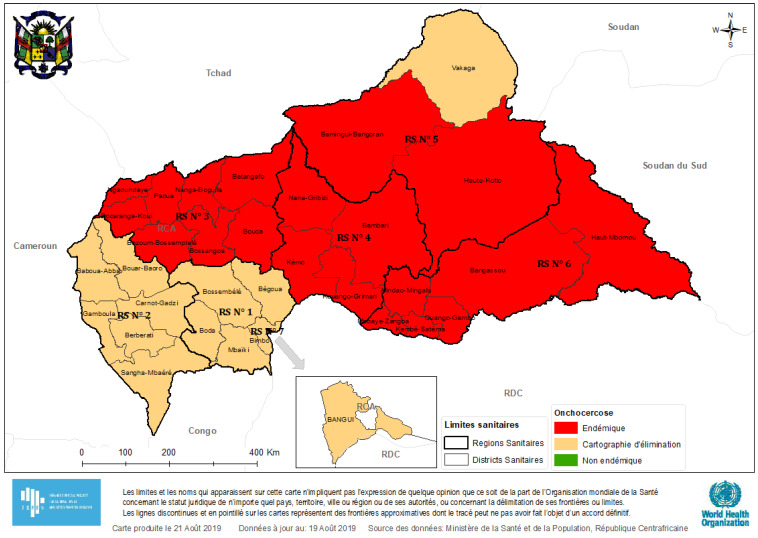
Onchocerciasis endemicity map of the Central African Republic, 2019.

**Figure 2 pathogens-09-00337-f002:**
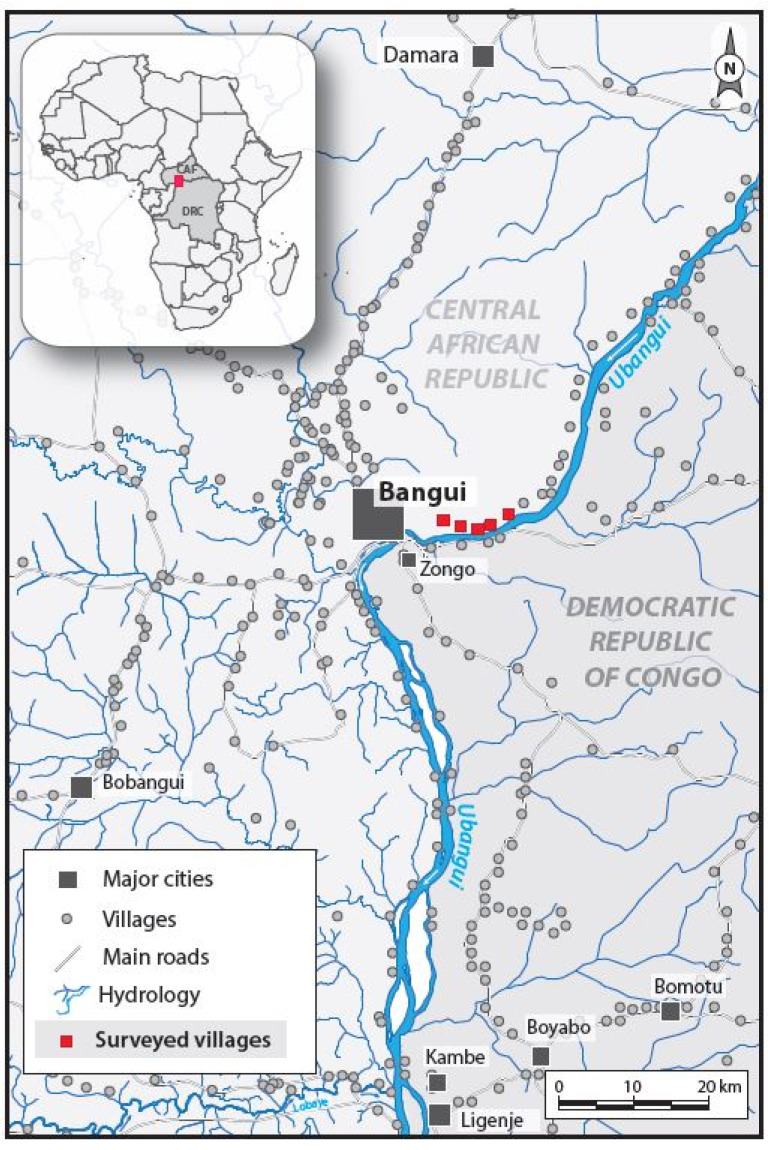
The villages of Landja Mboko included in the study.

**Table 1 pathogens-09-00337-t001:** Characteristics of study participants.

	Belespoir n = 30	Kodjo n = 50	Landja 1 n = 49	Landja 2 n = 106	Mangapou 2 n = 24
**Socio-demographics**					
Male: n (%)	19 (63.3)	25 (50.0)	20 (40.8)	58 (54.7)	13 (54.2)
Female: n (%)	11 (36.7)	25 (50.0)	29 (59.2)	48 (45.3)	11 (45.8)
Age: median (IQR)	7.0 (6.0–7.0)	7.5 (6.0–8.8)	7.0 (6.0–8.0)	7.5 (6.0–9.0)	7.0 (6.0–8.0)
Duration in village: median (IQR)	7.0 (6.0–7.0)	7.0 (6.0–8.0)	7.0 (6.0–8.0)	7.0 (6.0–8.0)	7.0 (6.0–8.0)
Enrolled in school: n (%)	30 (100.0)	43 (86.0)	43 (87.8)	90 (84.9)	20 (83.3)
School level: median (IQR)	1 (1–2)	2 (1–3)	1 (1–2)	2 (1–2)	2 (1–2)
**Anthropometrics**					
Height in cm: median (IQR)	113 (110-121)	112 (108–119)	114 (110–118)	119 (112–125)	112 (105–120)
HfA ^a^ z-score: mean (SD)	−1.04 (1.27)	−1.81 (1.51)	−1.12 (1.00)	−1.00 (1.38)	−2.25 (2.29)
Stunting ^b^: n (%)	6 (20.0)	22 (44.0)	8 (16.7)	27 (25.7)	13 (54.2)
Weight in kg: median (IQR)	20.5 (20.0–24.8)	20.0 (15.2–23.0)	22.0 (20.0–25.0)	22.0 (19.0–25.0)	20.5 (16.8–24.2)
BfA ^c^ z-score: mean (SD)	0.71 (0.80)	−0.67 (2.97)	0.60 (1.74)	−0.23 (1.71)	0.83 (2.06)
Underweight ^d^: n (%)	0 (0)	16 (32.0)	3 (6.3)	15 (14.4)	1 (4.2)

^a^ HfA: Height-for-age z-score based on the WHO growth curves; ^b^ Stunting: Height-for-age < −2z. ^c^ BfA: BMI-for-age z-score based on the WHO growth curves; ^d^ Underweight: BMI-for-age < −2z. IQR: Interquartile range; SD: Standard deviation.

**Table 2 pathogens-09-00337-t002:** Results of the rapid diagnostic tests.

	Belespoir	Kodjo	Landja 1	Landja 2	Mangapou 2
**Age-specific Ov16 results: n (%)**					
6 years	0 (0)	2 (10.0)	4 (17.4)	1 (2.8)	0 (0)
7 years	0 (0)	2 (40.0)	1 (7.7)	0 (0)	0 (0)
8 years	0 (0)	4 (33.3)	1 (11.1)	3 (13.6)	1 (14.3)
9 years	0 (0)	2 (15.4)	1 (25.0)	1 (3.2)	0 (0)
Overall ^a^	0 (0)	10 (20.0)	7 (14.3)	5 (4.7)	1 (4.2)
**Age-specific Lymphatic Filariasis (LF) results: n (%)**					
6 years	0 (0)	0 (0)	0 (0)	1 (2.8)	0 (0)
7 years	0 (0)	0 (0)	0 (0)	0 (0)	0 (0)
8 years	0 (0)	0 (0)	0 (0)	2 (9.5)	0 (0)
9 years	0 (0)	1 (7.7)	0 (0)	1 (3.2)	0 (0)
Overall ^b^	0 (0)	1 (2.0)	0 (0)	4 (3.8)	0 (0)

^a^ Overall Ov16 seroprevalence: 23/259 (8.9%); Difference across villages (Fisher exact test): *p* = 0.004. ^b^ Overall LF seroprevalence: 5/258 (1.9%); Difference across villages (Fisher exact test): *p* = 0.739.

**Table 3 pathogens-09-00337-t003:** Comparison of Ov16-positive and Ov16-negative participants.

	Ov16 Negative n = 236	Ov16 Positive n = 23	*p*-Value
Age: median (IQR)	7.0 (6.0–8.0)	8.0 (6.0–8.0)	0.378 ^a^
Male gender: n (%)	125 (53.0)	10 (43.5)	0.515 ^b^
School level: median (IQR)	2 (1–2)	2 (1–3)	0.569 ^a^
Stunting: n (%)	67 (28.5)	9 (40.9)	0.330 ^b^
Underweight: n (%)	28 (12.0)	7 (31.8)	0.018 ^b^
Itching: n (%)	83 (35.9)	14 (60.9)	0.034 ^b^
LF seropositivity: n (%)	3 (1.3)	2 (9.1)	0.059 ^c^

^a^ Mann–Whitney U test; ^b^ Chi-squared test; ^c^ Fisher exact test. LF: Lymphatic filariasis. IQR: Interquartile range.

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
