# Peer review of "Focus of Ongoing Onchocerciasis Transmission Close to Bangui, Central African Republic"

_pathogens, 2020, doi:10.3390/pathogens9050337_

Round 1

Reviewer 1 Report

Over all a well written paper reporting O. volvulus transmission in a low/non-endemic area. It highlights the importance of following up ancedotal reports.  I have a number of comments that need to be addressed/clarified

i) Introduction - need to include something on LF and filariasis co-endemicity, so the when the LF results are presented, it is not a surprise

ii) Methods - it is mentioned that the biplex test is used - can the authors be more expliciate in  

iii) What is the rational for sampling non-epiletic children and why aged betwee 6-9 and not other ages such as 5-10 or 5-7 years?  This may be helpful for the non-expert reader.      

iv) Could environmental change and/or population movement be responsible for change in transmission? Or was it previously poor or missed sampling?  It would be helpful to know if there were potential factors driving an increase in transmisison.

v) The lat/long coordinates of the villages should be included

vi) There are some inconsistencies in how the OV16 test and LF/lymphatic filariasis are refered to

vii) in the discussion Kodjo is noted to be 200m from the river Oubangui, it woudl be infomtative to knwo how far the other villages are from the river

viii) the conclusion states that more indepth re-assessment of the situation is warranted. Given the Loa co-endemicity, it owudl be useful to also asses the prevalence and coinfection as well as it may help to direct alternative strategies.  

Author Response

Over all a well written paper reporting O. volvulus transmission in a low/non-endemic area. It highlights the importance of following up anecdotal reports.  I have a number of comments that need to be addressed/clarified

  1. i) Introduction - need to include something on LF and filariasis co-endemicity, so the when the LF results are presented, it is not a surprise

Response

We now include in the introduction "The CAR is also considered to be endemic for lymphatic filariasis (LF) but there are no recent data about the prevalence of LF in the different parts of the country."

https://www.who.int/lymphatic_filariasis/resources/who_wer9344/en/

  1. ii) Methods - it is mentioned that the biplex test is used - can the authors be more expliciate in  

Response

The biplex test was used because an the OV16 RTD test was out of stock and could not be delivered in time to carry out the study. An additional advantage of the biplex test is that it also determines LF antibodies.

iii) What is the rational for sampling non-epileptic children and why aged between 6-9 and not other ages such as 5-10 or 5-7 years?  This may be helpful for the non-expert reader.      

Response

Sampling the 6-9 year old age groups is recommended by WHO to assess whether there is ongoing O. volvulus transmission.

World Health Organization. Report of the Second Meeting of the WHO Onchocerciasis Technical Advisory Subgroup. Geneva, Switzerland; 2018. Available from: https://apps.who.int/iris/bitstream/handle/10665/277238/WHO-CDS-NTD-PCT-2018.11-eng.pdf?ua=1.

We used this age group in several other studies in onchocerciasis endemic areas. This allows us also to compare data between study sites.

  1. iv) Could environmental change and/or population movement be responsible for change in transmission? Or was it previously poor or missed sampling?  It would be helpful to know if there were potential factors driving an increase in transmisison.

Response

We are not sure whether the onchocerciasis situation changed a lot at the study site. Indeed, it may be that in the past the onchocerciasis situations was not very well investigated. However it is also possible that in recent years onchocerciasis transmission increased because of an increasing number of people choose to live and farm close to the river sites because of the presence of fertile grounds. This river however contains blackfly breeding sites.

  1. v) The lat/long coordinates of the villages should be included

Response

We do not have the GPS data of the villages where the OV16 testing was done because of the lack of internet connection

  1. vi) There are some inconsistencies in how the OV16 test and LF/lymphatic filariasis are refered to

Response

We corrected the inconsistencies

vii) in the discussion Kodjo is noted to be 200m from the river Oubangui, it would be infomative to know how far the other villages are from the river

Response

The other villages were located about 1-3km from the river. We now mention this in the paper.

viii) the conclusion states that more in depth re-assessment of the situation is warranted. Given the Loa co-endemicity, it would be useful to also asses the prevalence and coinfection as well as it may help to direct alternative strategies.  

Response

We agree the determining the exact level of Loa-loa endemicity is important to decide about ivermectin use in the area. However such an assessment will have to be done at a later stage because of the current COVID-epidemic this is currently not possible.

We now include in in the conclusion: “Given the risk of encephalopathy when introducing ivermectin mass drug administration in loiasis endemic areas, the prevalence and intensity of L. loa infection also needs to be assessed in these villages.”

Reviewer 2 Report

This is a nicely written account of a very simple 'experiment' - looking at the seroprevalence (by a rapid diagnostic test) of antibodies to a recombinant antigen of onchocerciasis in children in the Central African Republic and comparing that with other developmental measures.  The finding of positive OV16 antibodies does imply the presence of onchocerciasis and identifies an area of CAR never before treated for oncho.  The oncho in this area will have to be addressed in order to achieve the revised global goal of complete elimination of oncho.  The problem, as the authors indicate, is that there is coexisting Loa infection in this area, so the specific management approach is not as simple as just administering ivermectin.  

Th manuscript's most useful fact that oncho is prevalent in this area of CAR is of value in a public health sense, but there is not too much more here than a well conducted simple study using an RDT that has not been as utilized as widely as it deserves to be.

Author Response

This is a nicely written account of a very simple 'experiment' - looking at the seroprevalence (by a rapid diagnostic test) of antibodies to a recombinant antigen of onchocerciasis in children in the Central African Republic and comparing that with other developmental measures.  The finding of positive OV16 antibodies does imply the presence of onchocerciasis and identifies an area of CAR never before treated for oncho.  The oncho in this area will have to be addressed in order to achieve the revised global goal of complete elimination of oncho.  The problem, as the authors indicate, is that there is coexisting Loa infection in this area, so the specific management approach is not as simple as just administering ivermectin.  

The manuscript's most useful fact that oncho is prevalent in this area of CAR is of value in a public health sense, but there is not too much more here than a well conducted simple study using an RDT that has not been as utilized as widely as it deserves to be

Response

We thank the reviewer for the kind remarks. The plan is to perform additional studies in the area ant better describe the situation in the area but given de COVID-19 epidemic these investigations for the moment are on hold.
